# The Role of Lactylation in Virus–Host Interactions

**DOI:** 10.3390/ijms26146613

**Published:** 2025-07-10

**Authors:** Gejie Zhao, Jia Zhou, Shutong He, Xiao Fei, Guijie Guo

**Affiliations:** 1Key Laboratory of Animal Pathogen Infection and Immunology of Fujian Province, College of Animal Sciences, Fujian Agriculture and Forestry University, Fuzhou 350002, China; 62306023020@fafu.edu.cn (G.Z.); 12306008032@fafu.edu.cn (J.Z.);; 2Fujian Province Joint Laboratory of Animal Pathogen Prevention and Control of the “Belt and Road”, College of Animal Sciences, Fujian Agriculture and Forestry University, Fuzhou 350002, China; 3Key Laboratory of Fujian-Taiwan Animal Pathogen Biology, College of Animal Sciences, Fujian Agriculture and Forestry University, Fuzhou 350002, China

**Keywords:** post-translational modifications (PTMs), lactylation, viral infection, innate immunity, virus

## Abstract

Lactylation, a novel form of post-translational modifications (PTMs) of protein, particularly within histone proteins, has recently gained attention for its role in regulating gene expression and cellular processes. In recent years, lactylation has been widely studied in cancer, immune diseases, neurological diseases, cardiovascular diseases, metabolic diseases, etc. Increasing evidence now suggests that lactylation also plays a significant role in the host’s innate immune response to viruses. Lactylation influences fundamental cellular functions, including transcriptional regulation, signal transduction, cell proliferation and differentiation. It affects protein behavior by modulating their function, stability, subcellular localization and interactions. Studies have shown that many viral infections promote lactate production through enhanced glycolysis, a process that facilitates viral replication. Given that innate immunity serves as the host’s first line of defense against pathogenic invasion, understanding how lactylation regulates antiviral responses offers promising avenues for the development of diagnostic tools and therapeutic strategies against viral diseases. In this review, we provide a comprehensive overview of recent research on the role of lactylation in viral–host interactions.

## 1. Introduction

Protein post-translational modifications (PTMs) have become a focal point in immunological research due to their pivotal role in regulating virus–host interactions. Common PTMs—including acetylation, methylation, S-nitrosylation, phosphorylation, ubiquitination, and lipidation—affect a wide range of physiological and pathological cellular processes. These modifications are integral to cellular function and thus hold significant value in the diagnosis, treatment, and prevention of various diseases. As a new type of PTMs, lactylation plays an importaant role in tumorigenesis, sepsis and immune disease development [1]. It represents a key pathway through which lactate exerts its biological functions. Lactylation is involved in several critical processes, including glycolysis-dependent cellular activity, macrophage polarization, vascular function, mitochondrial regulation, and nervous system homeostasis [2,3].

Viral infections often reprogram host cell metabolism, frequently enhancing glycolysis and thereby increasing lactate production. This metabolic shift can, in turn, influence the extent and function of protein lactylation. Studies have shown that several viruses, such as HIV, IAV, and SARS-CoV-2, can significantly reprogram host cell metabolism, inducing a “Warburg effect” that increases lactate production and creates a microenvironment conducive to viral replication [4,5,6]. Virus-induced lactate accumulation can lead to the lactylation of key host proteins or viral proteins. Lactylation is a novel bridge between cellular metabolism, epigenetic regulation, signaling and immune response. In this review, we focus on the impact and its underlying mechanisms of protein lactylation on both host immunity and viral replication during infection. Understanding these interactions may provide novel insights into the role of PTMs in viral pathogenesis and host defense.

## 2. Lactylation

Lactylation, a novel PTM, has been increasingly recognized for its role in various biological processes, including gene expression, cellular metabolism, and signaling pathways. This modification involves the covalent attachment of a lactic acid group to a protein’s lysine residue, primarily mediated by lactoyl coenzyme A (Lactoyl-CoA) [7].

### 2.1. Lactate and Lactylation

Lactate plays a critical role in animal physiology, serving not only as an essential energy source but also as a multifunctional signaling molecule. Glycolysis is one of the major pathways of metabolic pathways, through which cells extract energy. In this process, glucose is metabolized into pyruvate, yielding a small amount of ATP. However, under hypoxic conditions, pyruvate is not oxidized sufficiently, and the body maintains normal glycolysis by reducing pyruvate to lactate through the enzyme lactate dehydrogenase (LDH) (Figure 1). Beyond its metabolic role, lactate contributes to various physiological processes, including the suppression of lipolysis, modulation of immune responses, promotion of wound healing, and enhancement of gut microbial activity [8].

Lactate was first isolated from sour milk in 1780 by the Swedish chemist Carl Wilhelm Scheele [9]. For a long time, lactate was thought to be a metabolic waste product produced by cellular respiration under anaerobic conditions. In 1924, Otto Warburg famously observed that tumor cells preferentially utilize glycolysis to convert glucose into lactate even in the presence of oxygen—a phenomenon known as the “Warburg effect” [10]. The Warburg effect is not only present in tumors but also in a wide range of processes such as immunocyte activation and cellular reprogramming. For example, lactate can be taken up by tumor cells and transported to mitochondria for oxidation to provide energy, and lactate in the tumor microenvironment has an inhibitory effect on the killing function of immune cells [11]. The Warburg effect is closely related to molecular mechanisms, such as metabolic changes induced by the activation of oncogenes and the loss of tumor suppressor factors [12]. Importantly, metabolic labeling experiments using isotopically labeled L-lactate combined with mass spectrometry (MS/MS) analysis have demonstrated that lysine lactylation originates from lactate metabolism [13].

**Figure 1 ijms-26-06613-f001:**
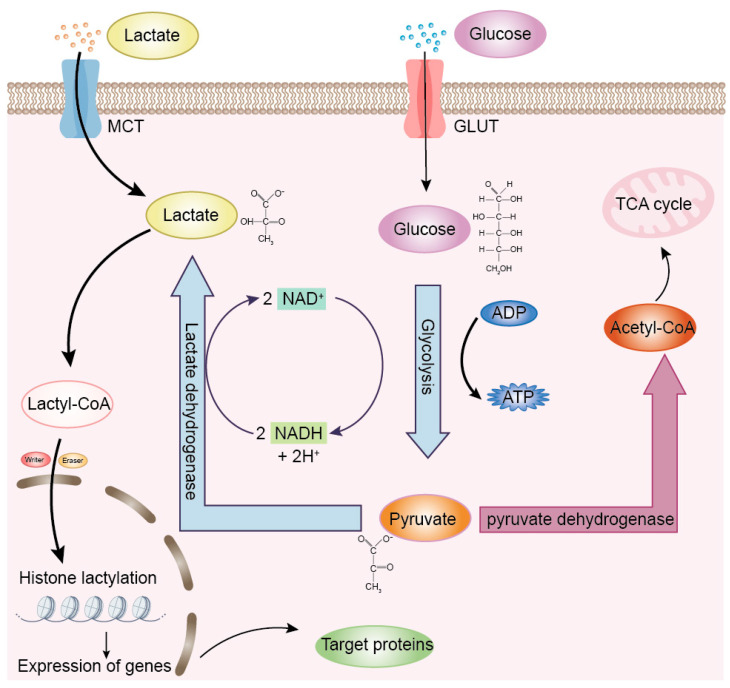
Overview of lactate production and its role in lactylation. Under aerobic conditions: Pyruvate is oxidized to acetyl-CoA by the pyruvate dehydrogenase complex (PDH). Acetyl-CoA then enters the tricarboxylic acid (TCA) cycle, followed by oxidative phosphorylation, ultimately generating ATP. Under anaerobic conditions: Pyruvate is reduced to lactate by the lactate dehydrogenase (LDH). Additionally, cells can import extracellular lactate via monocarboxylate transporters (MCTs). Intracellular lactate can be converted to lactyl-CoA, which serves as a substrate for lactylation modifications, thereby facilitating the lactylation of both histone and non-histone proteins. Inspired by the lactate metabolism framework described by Li et al. [14].

### 2.2. Discovery of Lactylation

In 2019, Yingming Zhao’s team at the University of Chicago reported for the first time a new epigenetic modification derived from lactate, lysine L-lactation (KL-la), and the existence of two isoforms, N-ε lysine (Kce) and D-lysine lactation (KD-la) [13]. Histone lactylation affects gene transcription and plays an important role in many biological processes, such as energy metabolism, tissue repair, cellular reprogramming, neural excitability, Alzheimer’s disease, and immunosuppression [15,16]. The results of modificationomics have revealed the existence of a very large number of non-histone proteins that undergo lactylation in addition to histones [17]. Gaffney et al. demonstrated that a wide array of non-histone proteins also undergo lactylation within various cellular compartments—including the nucleus, cytoplasm, mitochondria, endoplasmic reticulum, and plasma membrane [18]. These findings have overturned the traditional perception of lactate as a metabolic waste product and established its dual role as a signaling molecule and a modification donor.

### 2.3. Mechanisms and Regulation of Lactylation

Lactylation is a type of epigenetic modification that affects the structure and function of proteins by adding lactyl groups of lactate origin to specific protein lysine residues after the conversion of lactate to lactyl coenzyme A in the body [16]. Three known isomeric forms of lactylation (Kla) include KL-la, D-lysine lactylation (KD-la), and N-ε-lysine (carboxyethyl) lactylation (Kce) [19] (Figure 2). Lactylation plays important roles in the regulation of gene expression, protein function, and a variety of biological processes. Under normal physiological conditions, lactylation serves as a key metabolic–epigenetic bridge linking cellular metabolic states (e.g., glycolytic activity and lactate levels) to gene expression regulation. It promotes the transcription of specific genes by modifying histones and plays a central role in maintaining immune homeostasis and dynamically regulating the cell’s own metabolic pathways to adapt to physiological demands [13].

The regulation of histone lactylation can be categorized into three groups of proteins: Writers, Erasers, and Readers [15]. Writers are proteins that are responsible for adding modifying groups to histones, including modifications such as acetylation, methylation, and so on. Erasers are proteins that are responsible for removing modifying groups from histones, restoring them to their original state. Readers are proteins that are able to recognize and bind proteins with specific modifying groups, thereby regulating gene expression and other cellular processes. Together, these three classes of proteins are involved in the dynamic modification and functional regulation of histones. The process of lactylation is regulated by lactate transferase and delactase. Among the Writers, p300 was the first identified enzyme capable of catalyzing histone lactylation. It modulates gene expression related to macrophage polarization and immune activation [13]. In 2022, class I histone deacetylases (HDAC1–3) were identified as potent Erasers in vitro. Members of the sirtuin family (SIRT1–3) were also found to possess delactylation activity, marking the first systematic exploration of lactylation Erasers [20]. Using lactylation-targeted proteomics, Professor Long Zhang’s team discovered that alanyl-tRNA synthetases AARS1/2 are intracellular lactate sensors and function as lactate transferase. Notably, AARS2 was found to lactylate the DNA sensor Cyclic GMP-AMP Synthase (cGAS), thereby suppressing innate immune responses and facilitating viral immune evasion [21]. Independently, Fangfang Zhou’s group made similar discoveries, identifying AARS1 as a key lactate-binding protein that globally catalyzes lysine lactylation [22].

In addition to the presence of lactylation in histones, lactylation is also present within non-histone proteins. First, lactylation can lead to protein spatial disruption, conformational changes and charge neutralization, which in turn modulate protein function. For example, in Alzheimer’s disease, the lactylation of the amyloid precursor protein (APP) at the K612 site affects its metabolic pathway, inhibits the amyloid pathway, promotes the endosomal–lysosomal degradation pathway, and reduces the production of the pathological deposition of amyloid-β (Aβ) [23]. In addition, lactylation can affect protein stability; for example, in prostate cancer, the lactylation of hypoxia inducible factor-1α (HIF-1α) improves its stability in normoxia, which in turn promotes angiogenesis in prostate cancer [24]. Second, lactylation can affect the subcellular localization of proteins; for example, the lactylated high circulating levels of lactate and high mobility group box-1 (HMGB1) induce its nucleoplasmic translocation and regulate its distribution in the cell, mediated by p300/CBP [25]. In renal diseases, non-histone lactylation, such as mitochondrial fission 1 protein (Fis1) lysine 20 (Fis1 K20la) and acyl-CoA synthetase family member 2 (ACSF2) lysine 182 (ACSF2 K182la), targets key mitochondrial proteins, exacerbating abnormal mitochondrial division and disturbed energy metabolism [26,27]. Furthermore, lactylation modulates enzyme activities, including metabolic enzyme activities and DNA repair enzyme activities. The lactylation of some metabolic enzymes can regulate their activities, which in turn affects cellular metabolic processes; for example, hexokinase 2 (HK2)-Kla activates hepatic stellate cell pro-fibrotic genes [28]. In addition, the lactylation of DNA repair proteins can affect their repair function; for example, meiotic recombination 11 (MRE11) lysine 673 (MRE11 K673la) enhances homologous recombination repair and leads to chemoresistance [29].

## 3. Molecular Mechanisms of Lactylation in Viral Infections

Lactylation serves as a critical molecular interface between cellular metabolism and immune function during viral infections. It modulates key components of the innate immune response and directly influences the viral life cycle through host metabolic pathways and protein modifications. Emerging evidence from both eukaryotic and prokaryotic systems underscores the broad regulatory potential of lactylation in viral pathogenesis and host defense.

### 3.1. Modulation of Host Innate Immunity by Lactylation

Interferons (IFNs) are pleiotropic cytokines with antiviral, antitumor and immunomodulatory properties, which are central coordinators of immune responses and play an extremely important role in the immune system. There are mainly IFN-I, IFN-II and IFN-III. The mitochondrial antiviral signaling protein (MAVS) is a key stimulator of IFN-I, and studies have shown that during the activation of the RIG-I-like receptor (RLR) signaling pathway, glycolytic activity is typically suppressed—a process necessary for efficient IFN-I production. However, lactate has been shown to bind MAVS, impeding its oligomerization and mitochondrial localization, thereby disrupting RLR signaling and inhibiting IFN-I synthesis [11]. The relevant signaling pathways are shown in Figure 3.

cGAS is an intracellular immune monitor that recognizes abnormally occurring intracellular DNA, induces the downstream production of IFN signals, and activates the immune system [33]. Using human cytomegalovirus as a model, Professor Zhang Long’s team demonstrated a negative correlation between serum L-lactate levels and Cyclic GMP-AMPP (cGAMP) production in infected individuals, suggesting that lactate inhibits the activity of cGAS, and that cGAS lactylation inhibits the ability of DNA recognition and weakens immune surveillance [21]. The relevant signaling pathways are shown in Figure 3.

The nuclear factor kappa-B (NF-κB) signaling pathway regulates gene transcription in immune response, inflammation, cell differentiation and apoptosis [34]. In the nucleus, NF-κB binds to DNA and regulates the transcription of genes such as interferons, interleukins, and chemokines, which modulate inflammation, immune responses, and other related physiological processes [35]. Research suggests that infection with classical swine fever virus (CSFV) activates the NF-κB pathway by promoting the expression of H2B Kla and H2B K16la through the LDHA–lactate axis and that H2B K16la mediates p65 nuclear translocation through KPNA2, induces IFN-λ expression and inhibits CSFV replication [30]. The relevant signaling pathways are shown in Figure 3.

### 3.2. Effect of Lactylation on Viral Life Cycle

In the early stages of viral infection, the virus needs to fuse with the host cell membrane in order to enter the cell interior [36]. Lactylation reduces intracellular pH by increasing the glycolytic activity of host cells and producing large amounts of lactate [37]. For many enveloped viruses, membrane fusion occurs in endosomes, where the viral fusion proteins are activated under low pH conditions, and the enveloped virus enters the host cell through the fusion of the viral membrane with the cell membrane. For example, the F protein of SER viruses is activated at a low pH, which promotes the fusion of the virus with the host cell membrane [38]. The direct role of low pH due to lactate buildup in relation to membrane fusion is not clear, and we think this may be a new direction for research.

After the virus has successfully entered the host cell, its replication process is dependent on the host cell’s metabolic system [39]. It has been found that Sugarcane Mosaic Virus (SCMV) infection increases the activity of LDH and leads to intracellular lactate accumulation, which in turn affects the metabolic homeostasis of the host cell. It was found that LDH interacts with viral replication complexes (VRCs) and regulates the stability of VRCs through lactylation, thereby promoting viral replication [40].

L-lysine inhibits viral replication by competitively antagonizing arginine and interfering with viral nucleocapsid protein formation and DNA synthesis. This antagonistic effect of L-lysine is particularly important during viral infection. Importantly, this mechanism can function even during viral latency by blocking key stages such as viral adsorption, fusion, and protein synthesis [41].

Lactylation affects viral pathogenicity by influencing the viral life cycle. [42]. While most lactylation research has focused on eukaryotic systems, recent work has begun to explore its roles in prokaryotes. The team of Kingland Zhang and Liyan Xu reported for the first time the lactylation regulators in prokaryotes and further revealed a new mechanism of lactylation-mediated bacterial growth and energy metabolism [43]. Similarly, Min Li’s group demonstrated that Staphylococcus aureus modulates the lactylation of secreted proteins, including the key virulence factor α-toxin. This modification affects α-toxin’s receptor-independent membrane binding, oligomerization, and membrane integration, ultimately altering its pathogenic potential in host tissues [44]. This regulatory mechanism not only affects the pathogenicity of Staphylococcus aureus but may also provide a theoretical basis for the development of novel antimicrobial strategies against bacterial infections.

### 3.3. Lactylation as a Consequence of Viral Metabolic Reprogramming

Viruses have evolved sophisticated strategies to manipulate host cell metabolism to support their replication and spread [32]. One such strategy involves the alteration of host metabolic pathways, particularly glycolysis, to increase lactate production and provide more energy and metabolites for viral replication [31]. For example, Enterovirus A71 (EV-A71) infection provides a new target for viral muscle soreness (VMS) therapy by activating the HMGB1/LCN2/PDK1/lactate axis, causing muscle lactate accumulation and triggering muscle pain [45]. EV-A71 also activates the PI3K/Akt signaling pathway, which promotes host cell glycolysis to supply energy for viral replication and metabolize intermediates. Too low or too high glucose concentrations and excess lactate inhibit glycolysis and impair EV-A71 replication [46]. Similarly, lactonization-modified enzymes may accelerate the glycolysis process to provide more ATP and nucleotide precursors for viral replication, thus supporting efficient viral replication. This metabolic reprogramming is observed in various viral infections, including those caused by white spot syndrome virus (WSSV) and human cytomegalovirus (HCMV) [42,47].

In the case of WSSV infection in crustaceans, the virus induces an enhancement of glycolysis, leading to the accumulation of lactate. This lactate accumulation promotes site-specific histone lactylation, particularly H3K18la and H4K12la, which in turn upregulates the expression of genes such as ribosomal protein S6 kinases 2 (S6K2), thereby promoting viral replication and transmission. By targeting HIF-1α, viral miR-N20 modulates this process, suppressing host glycolysis and thereby reducing histone lactylation [42].

Similarly, HCMV infection in human cells also leads to an increase in glycolytic flux, resulting in substantial lactate accumulation. This lactate accumulation induces widespread protein lactylation, particularly in intrinsically disordered regions (IDRs) of proteins. The lactylation of viral proteins, such as pUL112, regulates viral protein–protein interactions and the formation of nuclear LLPS compartments, which are crucial for viral genome replication. Additionally, the lactylation of host immune signaling proteins, including RNA binding protein 14 (RBM14) and interferon-γ-inducible protein-16 (IFI16), promotes viral spread by subverting host immunity [47].

Furthermore, lactate production and secretion during HCMV infection contribute to the creation of a microenvironment that primes neighboring cells for infection. The secretion of lactate into the microenvironment may also function to temper any antiviral impacts of lactate accumulation within the infected cell.

In summary, viruses exploit host metabolic pathways, particularly glycolysis, to increase lactate production. This lactate accumulation induces widespread protein lactylation, which regulates viral replication, immune evasion, and cell-to-cell spread. Understanding these mechanisms is crucial for developing antiviral strategies that target the metabolic reprogramming induced by viruses.

## 4. Advances in Lactylation in Viral Infections

Research has shown that lactylation modifications are implicated in various pathogenic processes of viruses, including the viral reactivation, replication, propagation, and modulation of the host immune response [1,48,49]. These modifications can impact transcription, translation, and metabolic functions in host cells, thereby influencing the progression of diseases [7,50,51]. Lactylation has been shown to enhance the survival and proliferation of viruses by modulating metabolic pathways and immune responses in host cells. This mechanism is crucial for the viral life cycle and possesses substantial practical applications across numerous fields [7,52,53]. This section explores recent advancements in lactylation research, focusing on its role in DNA and RNA viruses. We discuss how lactylation influences viral pathogenesis and host immune responses, highlighting key findings and remaining gaps in knowledge (please refer to Table 1 for additional information).

**Table 1 ijms-26-06613-t001:** Recent advances in lactylation during virus–host interactions and their functional implications.

Virus	Virus Type	Host Cell Type	Related Proteins	Control Mechanism	Reference	Functionality
WSSV	DNA	Shrimp	S6K2 HIF-1α H3K18la H4K12la	Viral infection promotes glycolysis, leading to lactic acid accumulation. Promotes lactation of intracellular histones	[43]	Promotion of viral replication
HSV-1 KSHV MPXV	DNA	Human	ALKBH5 ESCO2 SIRT6	Viral infection enhances ALKBH5 lactylation and promotes IFN-β production	[36]	Inhibits viral replication
KSHV	DNA	Human	NAT10 ATAT1	NAT10 is lactylated by ATAT1 to promote viral reactivation	[17]	Promotion of viral replication
PRRSV	RNA	Swine	HSPA6 TRAF3 IKKε	Virus-induced lactylation activates HSPA6 expression and hinders IFN-β production	[48]	Promotion of viral replication
SFTSV	RNA	Human	YTHDF1 Sirt6 ESCO1	Viral infection increases YTHF1 lactylation and targets viral RNA for degradation	[54]	Promotion of viral replication
SVA	RNA	Swine	PKM PGK1 HIF-1α PDK3	Viral infection promotes lactate production, attenuates the interaction between MAVS and RIG-I, and inhibits IFN-β production	[31]	Promotion of viral replication
H1N1	RNA	Human	HK2 PKM2 PDK3 HIF-1	Induction of glycolytic pathway	[55]	Promotion of viral replication
CSFV	RNA	Swine	H2BK16la NS4A KPNA2	H2BK16la and pan Kla induce IFN-λ production through KPNA2-dependent activation of NF-κB pathway	[30]	Inhibition of viral replication

### 4.1. Lactylation in DNA Viruses

Virus-encoded microRNAs (miRNAs) can affect viral infection by modulating host metabolic processes, including lactylation. During infection by WSSV, the virion-associated compound palmitamide binds to triose-phosphate isomerase, thereby enhancing host glycolysis and facilitating viral replication [56]. WSSV infection upregulates glycolytic enzymes HK and LDH, leading to lactate accumulation that drives histone H3K18la/H4K12la lactylation—a key mechanism mediating virus–host interactions. This finding suggests that lactate, a metabolite of glycolysis, may play a key role in virus–host interactions [42].

Human Alk B homolog 5 (ALKBH5) plays a crucial role in the regulation of innate immunity. The lactylation of ALKBH5 attenuates viral replication. Mechanistically, lactylation of ALKBH5 is enhanced during Herpesvirus and Mpox virus (MPXV) infection by increasing the interactions of ALKBH5 with acetyltransferase establishment of sister chromatid cohesion N-acetyltransferase 2 (ESCO2) and decreasing the interactions of ALKBH5 with deacetyltransferase SIRT6. Emulsified ALKBH5 binds IFN-β mRNA, leading to the demethylation of its m6A modification and promotion of IFN-β production, attenuating viral replication [57]. This finding provides an understanding of innate immunity and offers a potential therapeutic target for herpes simplex virus-1 (HSV-1), Kaposi’s sarcoma-associated herpesvirus (KSHV), and MPXV infections.

In some cases, lactylation can also activate the host’s antiviral immune response. For example, N-Acetyltransferase 10 (NAT10) undergoes lactylation catalyzed by α-tubulin acetyltransferase 1 (ATAT1) during KSHV reactivation, which enhances its RNA acetyltransferase function and raises the level of N4-acetylcytidylic acid (ac 4 C) modification on the tRNA Ser-CGA-1-1, thus facilitating viral reactivation from latency [17].

### 4.2. Lactylation in RNA Viruses

It has been shown that porcine reproductive and respiratory syndrome virus (PRRSV) increases cellular lactate levels in a dose-dependent manner. PRRSV-induced lactate activates the expression of HSPA6. Mechanistically, lactate inhibits IFN-β induction by activating the lactate–lactylation–HSPA6 axis, thereby promoting PRRSV growth. HSPA6 reduces IFN-β induction by blocking the interaction between tumor necrosis factor receptor-associated factor 3 (TRAF3) and IκB kinase (IKK) epsilon (IKKɛ) [31]. This is the first study to examine the relationship between virus-induced lactylation and viral infection. The relevant signaling pathways are shown in Figure 3.

During infection with severe fever with thrombocytopenia syndrome virus (SFTSV), the viral protein NSs enhances the lactylation of the m^6^A reader protein YTH N6-Methyladenosine RNA Binding Protein F1 (YTHDF1). This modification inhibits YTHDF1’s ability to bind and degrade viral mRNAs, thereby enhancing viral replication. The study reveals a previously unrecognized interplay between m^6^A modification and viral immune evasion [54].

Senecavirus A (SVA)-induced glycolysis promotes viral replication by facilitating lactate production, which then attenuates the interaction between MAVS and RIG-I. The inhibition of RLR signaling improves SVA replication by promoting lactate production to attenuate the interaction between MAVS and RIG-I. Glycolysis regulates SVA replication mainly through RIG-I signaling [32]. The relevant signaling pathways are shown in Figure 3.

It was found that HK2 expression was up-regulated in influenza A virus (H1N1)-infected A549 cells, and viral infection activated the hypoxia-inducible factor 1 (HIF-1) signaling pathway. The inhibition of either glycolysis or HIF-1 significantly reduces viral replication, suggesting that influenza A virus induces the glycolytic pathway and thus promotes viral replication, but the level of viral replication after glycolytic inhibition or enhancement was not associated with IFN signaling [55].

### 4.3. Lactylation in Oncogenic Viruses

Oncogenic viruses, such as human papillomavirus (HPV) and hepatitis B virus (HBV), have been shown to manipulate lactylation to promote cancer progression [58,59]. Lactylation plays a pivotal role in cellular metabolism and is increasingly being recognized for its involvement in the development in tumorigenesis [13]. Recent studies have highlighted how lactate accumulation and protein lactylation within the tumor microenvironment enable cancer cells to evade immune surveillance and support their proliferation and survival [14,60].

In hepatocellular carcinoma (HCC)—a highly lethal form of liver cancer frequently associated with HBV and hepatitis C virus (HCV)—lactylation has emerged as a significant factor [7]. A study investigated the role of lactylation-related genes (LRGs) in HBV/HCV-associated HCC, identifying novel biomarkers for diagnosis and prognosis. Six LRGs—ALB, glucose-6-phosphate dehydrogenase (G6PD), HMGA1, MKI67, RACGAP1, and RFC4—were identified as potential molecular indicators of viral HCC. Immunohistochemistry (IHC) analyses of patient samples further validated that MKI67 and RACGAP1 were more strongly expressed in HBV/HCV-related HCC than in non-viral HCC. These findings suggest that MKI67 and RACGAP1 may serve not only as diagnostic markers but also as therapeutic targets for virus-induced HCC [58].

HPV, particularly high-risk types such as HPV16, plays a key role in the pathogenesis of cervical cancer. A study reported that HPV16 E6 promotes cancer cell proliferation by activating the pentose phosphate pathway (PPP). The study found that HPV16 E6 activates the PPP primarily by increasing G6PD enzyme activity. Mechanistically, E6 enhances the activity of G6PD—a rate-limiting enzyme in the PPP—by suppressing its lactylation. The inhibition of G6PD lactylation facilitates its dimerization, thereby increasing NADPH and glutathione (GSH) levels, reducing reactive oxygen species (ROS), and promoting cellular proliferation [59].

In conclusion, oncogenic viruses such as HPV and HBV manipulate host lactylation pathways to promote cancer progression. Emerging evidence supports the potential of lactylation-related genes (e.g., MKI67, RACGAP1) as biomarkers and therapeutic targets in virus-associated cancers [58]. Further research is needed to investigate the mechanisms underlying these interactions and to translate these insights into clinical interventions.

## 5. Conclusions and Future Perspectives

Lactylation, a newly identified post-translational modification, has emerged as a pivotal regulator in viral infections [13,21,42]. Recent research has highlighted its involvement in modulating host metabolic pathways and immune responses, thereby contributing to both viral survival and propagation [47,61,62].

Lactylation has been implicated in the pathogenesis of various viral diseases, including those caused by HBV, HHVs, HPV, MPXV, and SFTSV. For example, in SFTSV infection, the virus induces the lactylation of YTHDF1, suppressing the host’s antiviral response [54]. Conversely, during infections caused by HSV, KSHV, and MPXV, lactylation enhances the host’s innate immune response by modulating the functions of ALKBH5 and NAT10 [17]. Given these findings, therapeutic strategies aimed at modulating lactylation offer considerable potential. Interventions could include reducing intracellular lactate levels by inhibiting lactate synthesis or transport, thereby limiting the metabolic pathways that viruses exploit for replication [63,64]. Additionally, targeting lactylation-modifying enzymes, such as AARS1 and AARS2, may also prevent the modification of viral or host proteins, impairing key functions necessary for viral replication and immune evasion [21]. Furthermore, combining metabolic interventions with immunotherapeutic strategies holds potential for amplifying antiviral effects [65]. For example, employing metabolic modulators alongside checkpoint inhibitors might enhance the immune response against viruses [66,67]. However, such strategies must be carefully evaluated for systemic effects, as lactate is also essential for normal cellular function [68,69]. Therefore, personalized therapeutic approaches, tailored to the specific viral pathogen and host immune context, are essential [65].

In conclusion, the important role played by lactylation in protein transcriptional regulation involves several aspects of regulatory mechanisms. This novel type of post-translational modification plays a regulatory role in different aspects of energy metabolism, signaling and gene expression. Lactylation is closely linked to metabolic reprogramming, achieving a leap from metabolic phenotype to functional regulation.

Research into lactylation has not only broadened our understanding of host–virus dynamics but has also opened new avenues for the development of antiviral therapies. Future research should focus on identifying specific inhibitors and validating these strategies in a clinical setting to further exploit the therapeutic potential of lactylation in viral infections.

## Figures and Tables

**Figure 2 ijms-26-06613-f002:**
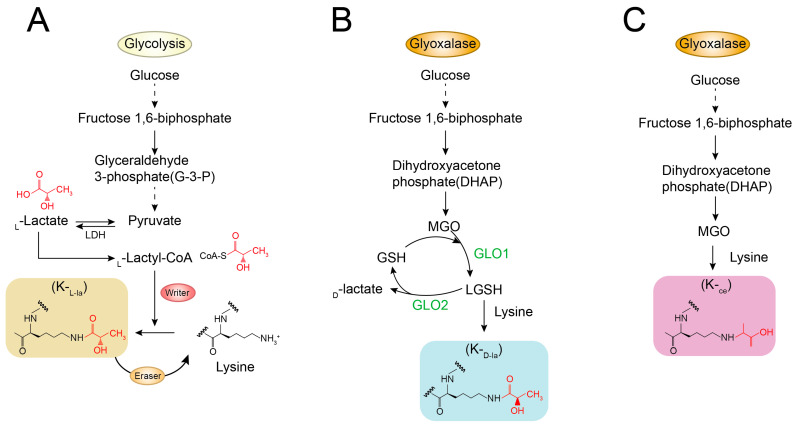
Three isomers of lysine lactylation. (**A**). KL-la: Forms enzymatically when glycolytically derived L-lactate (produced by LDH from pyruvate) is converted to lactyl-CoA and conjugated to lysine residues. (**B**). Lysine D-lactylation (KD-la): The stereoisomer of KL-la. Forms via spontaneous reaction between lysine residues and D-lactate generated by the glyoxalase pathway (processing the glycolytic byproduct methylglyoxal, MGO). (**C**). N-ε-(carboxyethyl)lysine (Kce): An adduct formed by the direct, non-enzymatic reaction of the highly reactive glycolytic byproduct methylglyoxal (MGO) with lysine residues. MGO is a highly reactive byproduct of glycolysis that can react with a variety of protein residues, including cysteine, arginine, and lysine. N-ε-(carboxyethyl)lysine (Kce), formed by the reaction with lysine, has been detected in cells, although its levels are lower than those of MGO-derived arginine modifications. Inspired by the lactylation isomer framework described by Zhang, D. et al. [19]. The dotted line indicates that the process involves multiple intermediate steps, which have been omitted here for simplicity. The solid line represents a direct reaction. Different colors are used solely for visual distinction.

**Figure 3 ijms-26-06613-f003:**
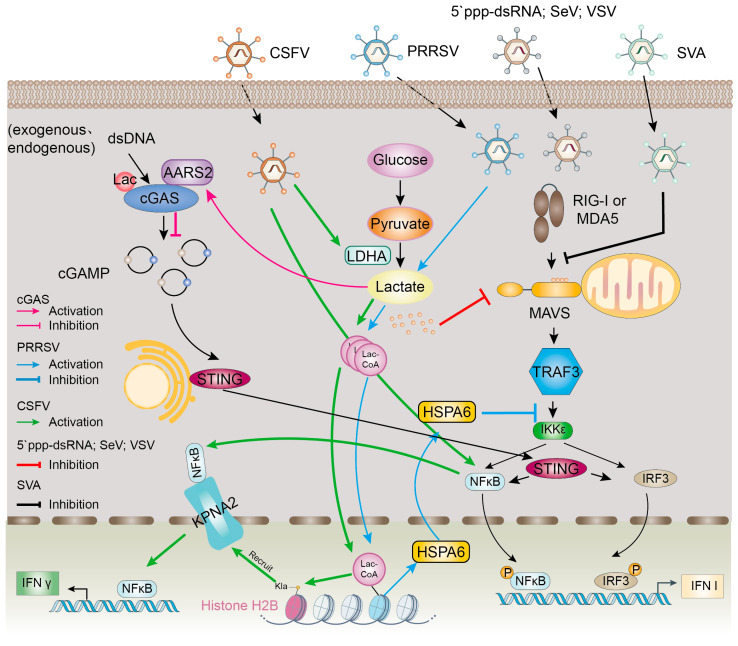
Lactylation regulates virus-associated signaling pathways. (1) cGAS Inhibition: Viral infection promotes the lactate-dependent lactylation of cGAS via the cGAS: AARS1/2 complex. This modification suppresses cGAS DNA-sensing activity, reducing cGAMP synthesis and inhibiting innate immunity. (2) PRRSV Immune Evasion: PRRSV infection elevates cellular lactate, driving the lactylation of Heat Shock Protein Family A Member 6 (HSPA6). Lactylated HSPA6 disrupts TRAF3-IKKε complex formation, blocking IFN-β production and compromising antiviral responses. (3) RLR Pathway Suppression: During infection by 5′ppp-dsRNA, SeV, or VSV, glycolytic lactate accumulation enables direct lactate binding to MAVS. This interaction prevents MAVS mitochondrial oligomerization, inhibiting RLR signaling activation. (4) CSFV Antiviral Defense: CSFV-induced lactate activates histone H2B lactylation (H2BK16la). This modification recruits karyopherin subunit alpha 2 (KPNA2) to facilitate p65/NF-κB nuclear translocation, stimulating IFN-λ expression and suppressing viral replication. (5) SVA Immune Evasion: Senecavirus A (SVA) enhances its replication by exploiting glycolysis-induced lactate production to disrupt the MAVS-RIG-I interaction and suppress RIG-I-like receptor (RLR) signaling. Inspired by the virus-associated signaling pathway framework described by Zhang, W., Li, H., Zhu, W., Pang, Y. et al. [11,21,30,31,32].

## Data Availability

No new data were created or analyzed in this study. Data sharing is not applicable to this article.

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
