# Peer review of "The Role of Lactylation in Virus–Host Interactions"

_ijms, 2025, doi:10.3390/ijms26146613_

Round 1

Reviewer 1 Report

Comments and Suggestions for Authors

Major points:

  1. Line 44-47: Added the missed references to the text.
  2. The role of the lactylation was not clarified in the normal conditions in line 111-112.
  3. The authors described the regulated mechanisms of histone by lactylation, how about the regulation of non-histone lactylation?
  4. Figure 3: It is better to show the “Modulation of host innate immunity by lactylation” of DNA virus and RNA virus, respectively.Furthermore, only DNA viruses are mentioned in the abstract, whereas the main text discusses more than just DNA viruses. Are the authors intending to focus solely on DNA viruses and lactylation? If so, the title may need to be revised for better accuracy.
  5. The lactate metabolic reprogramming induced by enterovirus, like EV-A71 (PMID 40378229,PMID 40127828) was also affect viral infection, however, this context was not discussed in your text.

Minor points:

  1. Line 54: Used the “PTM” to replace the “post-translational modification”.
  2. Line 110: What does the “Kla” mean? The full name of the word that first appears in the main text and the figure legends should be labeled.
  3. Please review the text carefully for the grammar errors, such as line 59, line 110, line 235,line 243,line 253.

Author Response

Response to Review 1:

General response:

Dear Reviewer,

Thank you very much for your insightful comments and professional suggestions regarding our manuscript entitled “The Role of Lactylation in Virus-Host Interactions.” Your feedback is highly valuable and has greatly helped us to improve and refine our work.

In response to your suggestions, we have thoroughly reviewed relevant literature and added more references to the revised manuscript. We have included additional discussion on the role of lactylation under normal conditions and addressed the mechanisms of non-histone lactylation. We also clarified the mention of “DNA virus” in the abstract and have added information on lactate metabolism reprogramming in enteroviruses such as EV-A71, as you recommended. Furthermore, for the first or second mention of relevant molecules, we have ensured full names or appropriate abbreviations are provided. We have also made a concerted effort to improve the grammar and clarity throughout the manuscript.

We have highlighted all changes using yellow markers in the revised manuscript. We sincerely thank you for your valuable suggestions and hope that the revisions meet your expectations.

Major points:

1. Line 44-47: Added the missed references to the text.

Response 1: Thank you for pointing this out. We have added the corresponding references in line 47.

2. The role of the lactylation was not clarified in the normal conditions in line 111-112.

Response 2: You are correct. Our original focus was primarily on the regulatory role of lactylation under pathological conditions, which was not comprehensive. We have revised and added relevant content in lines 111–116.

3. The authors described the regulated mechanisms of histone by lactylation, how about the regulation of non-histone lactylation?

Response 3: Agreed. We have, accordingly, added content related to the regulatory mechanisms of non-histone lactylation in lines 149-171.

4. Figure 3: It is better to show the “Modulation of host innate immunity by lactylation” of DNA virus and RNA virus, respectively.Furthermore, only DNA viruses are mentioned in the abstract, whereas the main text discusses more than just DNA viruses. Are the authors intending to focus solely on DNA viruses and lactylation? If so, the title may need to be revised for better accuracy.

Response 4: Thank you. In Figure 3, we illustrate the innate immune pathways affected by lactylation, particularly focusing on RNA viruses, due to the limited reports on DNA virus-mediated lactylation. Additionally, we have revised the abstract in line 17 to reflect our broader focus beyond DNA viruses.

5. The lactate metabolic reprogramming induced by enterovirus, like EV-A71 (PMID 40378229,PMID 40127828) was also affect viral infection, however, this context was not discussed in your text.

Response 5: We agree with your suggestion and have added a discussion on lactate metabolic reprogramming in EV-A71 in lines 259–265.

Minor points:

1. Line 54: Used the “PTM” to replace the “post-translational modification”.

Response 1: Thank you. We have replaced the phrase with "PTM" in line 54.

2. Line 110: What does the “Kla” mean? The full name of the word that first appears in the main text and the figure legends should be labeled.

Response 2: Thank you for pointing this out. "Kla" refers to lysine lactylation, and we have defined it in line 109.

3. Please review the text carefully for the grammar errors, such as line 59, line 110, line 235,line 243,line 253.

Response 3: Thank you for the reminder. We have made the necessary corrections. For example, in line 58, "lactate acid" has been corrected to "lactate". We also refined language in lines 274–275 and reviewed the entire manuscript for other grammatical issues.

Reviewer 2 Report

Comments and Suggestions for Authors

This is a very interesting, very informative review of an important topic.

There are some errors in some of the details discussed in the review; these errors should be corrected and the authors should be more careful to avoid errors in the future.

1.The legend to Fig. 1 refers to White et al (57). However, ref.57 is by Li et al. I have no idea how to find the article by White et al.

2.Line 96 contains the word “exerts” but this is not the correct word here and should be replaced.

3.Line 120 states that “cellular levels are lower…” but should be specific: cellular levels of what?

4.There are numerous references to HSPA6, but this is never identified. Similarly, what is KPNA2?

5.Lines 196-198 make the very interesting claim that lactate acidifies the cellular environment, promoting fusion of the virus with the host cell membrane, citing Tyl et al (ref. 24). However, Tyl et al. say nothing about membrane fusion; it is certainly misleading to make this statement and attribute it to Tyl et al. I know of no evidence connecting lactate metabolism with membrane fusion. In general, when viruses rely on low pH to trigger membrane fusion, it is the low pH in vacuoles or endosomes, not the cytoplasm itself.

6.Line 298 refers to “the lactation-HSPA6 axis”. As noted above, I do not know what HSPA6 is, and in addition, do the authors mean lactylation rather than lactation?

7.Line 322 discusses “oncogenic viruses such as HPV and HBV”, citing ref. 43. But ref 43 is about HCV and HBV and does not mention HPV at all. This mistake must be corrected.

Author Response

Response to Review 2:

General response:

Dear Reviewer,

Thank you for your encouraging feedback and constructive comments on our manuscript. We have carefully considered each of your suggestions and made corresponding revisions, which are marked in blue in the revised manuscript.

We have corrected mismatches in figure legends and references, clarified terminology such as “cellular levels,” and revised incorrect or misleading statements. We also removed improper attributions and provided accurate references where necessary. We are grateful for your detailed and thoughtful review, which has helped us significantly improve our manuscript.

1. The legend to Fig. 1 refers to White et al (57). However, ref.57 is by Li et al. I have no idea how to find the article by White et al.

Response 1: Thank you. This was an oversight. We have corrected the citation in line 90 to reflect the correct reference, Li et al.

2. Line 96 contains the word “exerts” but this is not the correct word here and should be replaced.

Response 2: We sincerely thank the reviewer for careful reading as suggested by the reviewer, we have corrected the “exerts” into “affects” in line 95.

3. Line 120 states that “cellular levels are lower…” but should be specific: cellular levels of what?

Response 3: Thank you for pointing this out. MGO is highly reactive and is known to react with several protein residues, including cysteine, arginine, and lysine. MGO reacts with the ε - amino group of lysine residues to form Kce. Although Kce can be detected in cells, its levels are usually much lower than those of MGO - derived arginine modifications such as MG - H1, because these modifications are more stable and more common in cells. We have added this in the text in lines 122-128.

4. There are numerous references to HSPA6, but this is never identified. Similarly, what is KPNA2?

Response 4: Thank you for pointing this out. We have added the full names of HSPA6 and KPNA2 in lines 194 and 199.

5. Lines 196-198 make the very interesting claim that lactate acidifies the cellular environment, promoting fusion of the virus with the host cell membrane, citing Tyl et al (ref. 24). However, Tyl et al. say nothing about membrane fusion; it is certainly misleading to make this statement and attribute it to Tyl et al. I know of no evidence connecting lactate metabolism with membrane fusion. In general, when viruses rely on low pH to trigger membrane fusion, it is the low pH in vacuoles or endosomes, not the cytoplasm itself.

Response 5: We sincerely apologize. We misinterpreted the literature. In reading the literature, we found that lactate buildup leads to a decrease in pH levels, and it has been shown that many viral fusion proteins are activated in low pH environments, resulting in a conformational change that promotes fusion of the viral envelope with the endosomal membrane. We have previously drawn the wrong conclusion by arbitrarily attributing this low pH-triggered membrane fusion directly to lactate buildup. Certainly, we believe that the potential connection between the two may be a subject worthy of further study. We have made changes, in lines 224-231.

6. Line 298 refers to “the lactation-HSPA6 axis”. As noted above, I do not know what HSPA6 is, and in addition, do the authors mean lactylation rather than lactation?

Response 6: Thank you for the correction. We have revised this to “Lactate–lactylation–HSPA6 axis” to reflect the intended meaning, consistent with line 336.

7. Line 322 discusses “oncogenic viruses such as HPV and HBV”, citing ref. 43. But ref 43 is about HCV and HBV and does not mention HPV at all. This mistake must be corrected.

Response 7: Thank you for pointing this out. This is a general statement and does not imply that it is from the same reference. It was an oversight on our part, and we have added new references to support this statement in lines 360-361.

Reviewer 3 Report

Comments and Suggestions for Authors

This review article aims to provide a comprehensive overview of the emerging role of lactylation in virus-host interactions, discussing its mechanisms, regulatory pathways, and implications in both DNA and RNA virus infections.

Many sections of the manuscript, particularly those discussing DNA and RNA viruses, are overly general and lack specificity. These sections fail to provide sufficient mechanistic detail or meaningful comparative analysis between different viral systems.

Overall, the discussion remains superficial and lacks the depth. A more thorough and critical examination of virus-specific pathways, supported by recent experimental evidence, is needed to enhance the manuscript’s scientific value.

In many places, the language appears to be AI-assisted (e.g., ChatGPT). If such tools were used, the authors must take responsibility for rigorous editing and refinement.

Author Response

Response to Review 3:

General response:

Dear Reviewer,

We sincerely thank you for your thoughtful and detailed review. We carefully considered your comments and have addressed them below. We acknowledge the overgeneralization in parts of our discussion on DNA and RNA viruses and the need for greater specificity. However, we currently face several limitations:

1. Data and Literature Limitations:

There is a lack of available data and literature providing detailed mechanistic insights into virus-specific lactylation pathways. Nevertheless, we have included as much mechanistic detail as current research allows.

2. Scope and Space Constraints:

Our aim was to provide a broad overview of lactylation in virus-host interactions. A more detailed discussion of each viral system may exceed the scope of this review but will be considered in future publications.

3. Emerging Nature of the Field:

Lactylation in virus-host interatctions is a newly emerging research topic. Many mechanisms are still under investigation. We have cited the most recent and relevant studies to ensure that our discussion reflects current knowledge and highlights areas for future research.

Regarding your concerns about language, we have re-edited the manuscript carefully to improve clarity, coherence, and scientific tone.

1. Many sections of the manuscript, particularly those discussing DNA and RNA viruses, are overly general and lack specificity. These sections fail to provide sufficient mechanistic detail or meaningful comparative analysis between different viral systems.

Response 1: Thank you. Due to space limitations, we provided a macroscopic overview to highlight the significance of lactylation in virus-host interactions. We added a specific example of EV-A71-induced lactate metabolic reprogramming to enhance depth.

2. Overall, the discussion remains superficial and lacks the depth. A more thorough and critical examination of virus-specific pathways, supported by recent experimental evidence, is needed to enhance the manuscript’s scientific value.

Response 2: We appreciate your thorough review of our manuscript. Due to the limited experimental data and literature resources currently available, we have discussed the role of lactylation in virus-host interactions in as much detail as possible with the resources available. Lactylation is an emerging area of research and many of the mechanisms are not yet fully understood. We have cited the most recent research literature to ensure that our discussion is based on the latest scientific findings. However, due to the rapid progress of research in this field, some specific mechanistic details may require further studies to validate. We add that lactylation leads to a decrease in intracellular pH by enhancing the glycolytic activity of the host cell, which in turn produces large amounts of lactate, and we elaborate that for many enveloped viruses, the process of membrane fusion is dependent on a low pH environment, which facilitates the fusion of the virus with the host cell membrane in lines 224-231.

3. In many places, the language appears to be AI-assisted (e.g., ChatGPT). If such tools were used, the authors must take responsibility for rigorous editing and refinement.

Response 3: Thank you for the reminder. We have carefully proofread and polished the manuscript to ensure academic quality in language and tone.

Round 2

Reviewer 3 Report

Comments and Suggestions for Authors

Authors have addressed my concerns. However, it would have been better to add more details.